# PSHead: 3D Head Reconstruction from a Single Image with Diffusion Prior and Self-Enhancement

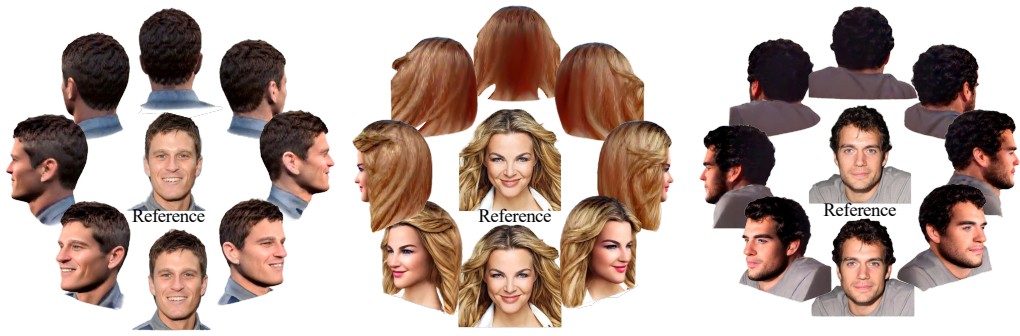

"a [V] man, brown short hair, detailed strand hair, gray polo shirt, photorealistic, 8K, HDR"

"a [V] woman, blonde long hair, detailed strand hair, white shirt, photorealistic, 8K, HDR"

"a [V] man, brown curly hair, detailed strand hair, gray t shirt, photorealistic, 8K, HDR"

Figure 1: Conditioned on text and a reference image (in middle), our PSHead, can automatically generate a high-fidelity facial avatar. Each avatar is rendered from eight distinct viewpoints.

## Abstract

In this work, we investigate the problem of creating high-fidelity photorealistic 3D avatars from only a single face image. This task is inherently challenging due to the limited 3D cues and ambiguities present in a single viewpoint, further complicated by the intricate details of the human face (*e.g.*, wrinkles, facial hair). To address these challenges, we introduce PSHead, a coarse-to-fine framework that optimizes 3D Gaussian Splatting for a single image, guided by a mixture of object and face prior to generate high-quality 3D avatars while preserving faithfulness to the original image. At the coarse stage, we leverage diffusion models trained on general objects to predict coarse representation by applying score distillation sampling losses at novel views. This marks the first attempt to integrate text-to-image, image-to-image, and text-to-video diffusion priors, ensuring consistency across multiple views and robustness to variations in face size. In the fine stage, we utilize pretrained face generation models to denoise the rendered noisy images, and use them as supervision to refine the 3D representation. Our method outperforms existing approaches on in-the-wild images, proving its robustness and ability to capture intricate details without the need for extensive 3D supervision.

## 1 Introduction

Creating photorealistic 3D avatars is a key challenge in computer graphics, with applications in movies, games, virtual or augmented reality, and the metaverse. There is growing interest in creating digital avatars from a single image, as it is easily obtainable. While humans can intuitively infer 3D shapes and textures from a quick glance, thanks to their vast knowledge of the natural world, tackling this task algorithmically is far more difficult. The main challenge lies in the limited 3D cues and inherent ambiguities present in a single viewpoint, compounded by the rich and intricate details of the human face (*e.g.*, wrinkle, facial hair), making the task even more difficult.

Some attempts have been made to generate 3D heads from a single reference image, but their performance and flexibility are severely constrained by the training datasets. They typically utilize small

scale 3D head datasets (Zheng et al., 2024; Chen et al., 2024a) or large-scale 2D images (Chan et al., 2022a; An et al., 2023). However, challenges in capturing and processing data often result in reduced quality and diversity (in terms of identity, race, age *et al.*) in the training datasets, which in turn negatively impacts the accuracy of generated reconstructions, particularly when the reference image is captured in the wild. Additionally, the normalization preprocessing steps (*e.g.*, align-cropping) in (Chan et al., 2022a; An et al., 2023) prohibit them from handling inputs of varying scales, such as head only, head and neck, or head and shoulders images.

Recently, significant progress has been made in text and image-to-3D object generation, largely driven by diffusion models pretrained on large-scale datasets that encode object priors (Saharia et al., 2022; Rombach et al., 2022). The typical approach involves optimizing a 3D representation by aligning its 2D renderings from random angles with diffusion prior (Poole et al., 2023). While these methods have been successfully applied to text-to-3D avatar generation (Cao et al., 2024; Han et al., 2024; Liu et al., 2024), adapting them for image-to-3D avatar generation is non-trivial and requires additional efforts. The primary difficulty lies in achieving fidelity: the generated 3D models must closely match the identity of the reference image, while also being realistic at the same time, rather than relying on the more general guidance of a rough text prompt. However, most existing text-to-image (T2I) (Ruiz et al., 2023), image-to-image (I2I) (Liu et al., 2023), and text-to-video (T2V) (Wang et al., 2023b) diffusion models are not specifically trained on face images, limiting their ability to capture fine facial details or maintain identity-preserving characteristics. Face-specific diffusion models often have limitations: they either lack scale (He et al., 2024), rely on synthesized data (Wang et al., 2023a), or focus solely on frontal or profile views with facial landmark constraints (CrucibleAI, 2023), and are therefore not directly applicable to image-to-3D avatar task. Despite these limitations, Text-to-3D avatar generation under T2I guidance has demonstrated a strong ability to handle a wide range of inputs, from close-up shots of the face (Han et al., 2024; Liu et al., 2024) to full-body characters (Cao et al., 2024). This suggests that diffusion models possess valuable 3D knowledge about the human structure. Motivated by this, we argue that carefully leveraging existing diffusion models holds strong potential to solve the image-to-3D avatar task.

In this work, we propose a head-specific generative method PSHead that lifts a single frontal face image to an accurate and faithful 3D gaussian splatting (3D-GS) reconstruction (Kerbl et al., 2023), with a particular focus on preserving the subject's identity and recovering details in the reference image (*e.g.*, face, hair, neck, and shoulders). We adopt a coarse-to-fine strategy. At the coarse stage, we incorporate Score Distillation Sampling (SDS) (Poole et al., 2023) guidance from multiple types of diffusion models to leverage their unique strengths. These models include a subject-specific T2I model finetuned via DreamBooth (Ruiz et al., 2023), capturing person-specific characteristics such as hair style; an I2I model (Liu et al., 2023) to generate novel views with camera rotations covering a full 360° space and the reference image, providing plausible multiview SDS guidance; and a T2V model (Wang et al., 2023b) to generate novel views as consecutive frames in video, enhancing multiview consistency via a temporal cross-attention mechanism. The combined SDS loss from object diffusion models allows us to learn a 3D-GS with coarse geometry and a noisy appearance, lacking high-quality details, especially in the face and hair. To address this, we incorporate a refinement stage, where additional 2D facial priors from models trained on face datasets are used to refine the representation and enhance facial detail. Specifically, we prioritize enhancement by using landmark-guided ControlNet (CrucibleAI, 2023) to denoise the entire face image, with particular focus on refining face geometry and applying a face super-resolution model (Zhou et al., 2022) to increase resolution in facial regions. Additionally, we use the personalized T2I model to effectively denoise rest views, ensuring consistency across different angles.

To summarize, we make the following contributions: **(1)** We propose PSHead, a method that learns a 360° photographic 3D-GS representation for a reference image with varying face sizes; **(2)** We leverage a mixture of diffusion priors to generate a coarse representation of the input face, providing insights into how each prior contributes to the process; **(3)** We refine the coarse representation in an innovative way by introducing 2D face priors to enhance more detailed representation.

## 2 RELATED WORK

**Text to 3D.** A common approach to optimizing a 3D representation for text description is to optimize its 2D rendered images with guidance from diffusion-based text-guided 2D image generation models (Saharia et al., 2022; Rombach et al., 2022). DreamFusion (Poole et al., 2023) pioneers in proposing a SDS strategy to self-optimize neural radiance fields (NeRF) (Mildenhall et al., 2020)

with Imagen (Saharia et al., 2022). To apply it to 3D avatar generation, geometry parametric priors are employed to guide the learning of avatar shapes. DreamAvatar (Cao et al., 2024) learns a SMPL-based (Bogo et al., 2016) NeRF (Mildenhall et al., 2020) to incorporate human shape prior, while Headsculpt Han et al. (2024) leverages FLAME (Li et al., 2017) via landmark-guided ControlNet (CrucibleAI, 2023) to capture facial shape prior. HeadArtist (Liu et al., 2024) addresses challenges like over-saturation and smoothing from SDS by introducing self-score distillation. Our method is inspired by these approaches and also utilizes landmark-guided ControlNet that encodes facial shape priors. However, rather than relying on a fixed shape template, which cannot account for individual facial differences, we estimate 3D landmarks dynamically during training. Using their projection to image space to provide shape input for landmark-guided ControlNet.

**Image to 3D.** Image to 3D task involves reconstructing a 3D model from a single image, which is particularly challenging due to its ill-posed nature. One straightforward solution, following the text-to-3D pipeline (Richardson et al., 2023; Chen et al., 2023; 2024c), is to add reference image reconstruction at a specific viewpoint, using SDS (Poole et al., 2023) from diffusion models (Saharia et al., 2022; Rombach et al., 2022) to guide the rendered images from random views.

RealFusion (Melas-Kyriazi et al., 2023) starts with model personalization by creating a textual inversion embedding for the input image, then optimizes InstantNGP (Müller et al., 2022) progressively from low-to-high resolutions. Make-it-3D (Tang et al., 2023)

| Method | T2I | I2I | T2V | Pers. |
|---|---|---|---|---|
| RealFusion (Melas-Kyriazi et al., 2023) | ✓ | ✗ | ✗ | ✓ |
| Make-it-3D (Tang et al., 2023) | ✓ | ✗ | ✗ | ✗ |
| Magic123 (Qian et al., 2024) | ✓ | ✓ | ✗ | ✓ |
| DreamGaussian (Tang et al., 2024) | ✓ | ✓ | ✗ | ✗ |
| Ours | ✓ | ✓ | ✓ | ✓ |

Table 1: Summary of four design properties contributing to the image-to-3D task. Pers. denotes personalization.

learns to create finely detailed textured point clouds from a coarse NeRF, guided by T2I diffusion. Magic123 (Qian et al., 2024) optimizes a high-resolution mesh and texture from NeRF outputs, leveraging both T2I and I2I diffusion models for guidance. DreamGaussian (Tang et al., 2024) focuses initially on optimizing a 3D-GS with I2I guidance then refines a textured mesh by denoising. While these methods have shown promising outcomes for general objects, their performance in creating avatars from front-view facial images is hindered by the absence of face priors. Moreover, T2V diffusion, essential for maintaining multi-view consistency as demonstrated in (Kwak et al., 2024), has yet to be employed or analyzed in this context. We identify four potential factors that contribute to the success of the image-to-3D task, which are summarized in Table 1. Our method stands out from previous works by thoroughly analyzing how each of these components contributes to learning an accurate 3D representation from a single face image.

**Single Face to 3D.** The task of converting a single face to a 3D model can be divided into two categories based on the usage of different face generation models. The first approach, 3D-GAN inversion, involves initially training a 3D-GAN on a large-scale 2D face dataset and then learning the latent code for a specific face image. EG3D (Chan et al., 2022b) exemplifies this method, with subsequent studies enhancing inversion performance through the integration of symmetry priors (Yin et al., 2023), refinements (Bhattarai et al., 2024), and other techniques (Trevithick et al., 2023). The second approach focuses on generating a 3D avatar starting from a text prompt, where a face image is generated using T2I diffusion models, and a 3D model is learned with supervision from the generated image and guidance from the diffusion model. This method tends to prioritize textual descriptions over the input image, producing a 3D model that aligns more closely with the text description (Wu et al., 2024). However, these methods often rely heavily on the preprocessing steps used during the training of the 3D-GAN, making it difficult to generalize to arbitrary facial inputs.

## 3 METHOD

Here, we introduce PSHead, a coarse-to-fine pipeline designed for high-fidelity 360° avatar generation from a single frontal face. We begin by presenting the preliminary knowledge in Sec. 3.1, followed by a detailed description of our proposed method PSHead in Sec. 3.2.

### 3.1 PRELIMINARIES

**3D Gaussian Splatting (3D-GS)** (Kerbl et al., 2023) represents a 3D scene using a set of Gaussian primitives, rendering images through volume splatting. Each Gaussian primitive is represented by

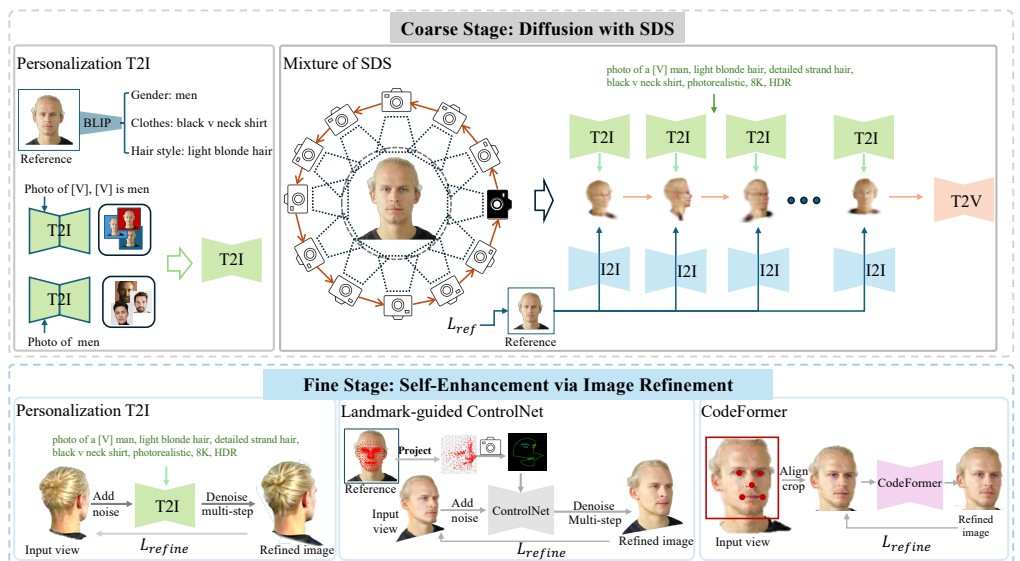

Figure 2: Overview of proposed PSHead method. Starting with a frontal view image as a reference, in the coarse stage, we first use DreamBooth to learn a personalization T2I diffusion. Then, using a combination of personalization T2I, I2I, and T2V diffusions, we apply a mixture of SDS on rendered novel view images to generate a coarse 3D-GS. In the fine stage, we enhance the 3D-GS by refining the image quality, supervised by the personalized T2I diffusion, landmark-guided ControlNet, and a pretrained face super-resolution model CodeFormer.

its position (mean) $\mu \in \mathbb{R}^3$, rotation $R \in \mathbb{R}^4$, scale $S \in \mathbb{R}^3$, view-dependent color as Spherical Harmonics coefficients $c \in \mathbb{R}^3$, and opacity value $\alpha \in \mathbb{R}$. Given a viewpoint $v$, the Gaussians can be rendered to the multi-channel image $\mathbf{I}$ through tile-based differentiable rasterization:

$$\mathbf{I} = \mathcal{R}(\mu, R, S, \alpha, c; v). \tag{1}$$

We use 3D-GS to represent the facial appearance and geometry because of its outstanding performance, flexibility, and real-time rendering efficiency, especially its ability to capture intricate details, such as hair strands, wrinkles and eyeglasses in human face (Chen et al., 2024b).

**Score Distillation Sampling (SDS)** introduced in DreamFusion (Poole et al., 2023), utilizes a pretrained diffusion model (Saharia et al., 2022) to validate multiple views of a given object. In our approach, we denote the optimizable parameters in 3D-GS as $\theta = \{\mu, R, S, \alpha, c\}$, its rendered image at random view as $\mathbf{I}$ and a pretrained diffusion model as $\phi$. We use SDS loss to optimize 3D-GS by performing gradient descent with respect to $\theta$ by:

$$\nabla_\theta \mathcal{L}_{SDS} = \mathbb{E}_{\epsilon,t}[w_t(\epsilon_\phi(\mathbf{I}_t) - \epsilon)\frac{\partial(\mathbf{I})}{\partial(\theta)}], \tag{2}$$

where $\epsilon$ is the Gaussian noise, $\mathbf{I}_t = \alpha_t \mathbf{I} + \sigma_t \epsilon$ is the noised image, $\alpha_t$, $\sigma_t$, and $w_t$ are noise sampler terms. Intuitively, Eq 2 measures the difference between the Gaussian noise $\epsilon$ added to the rendered image $\mathbf{I}$ and the predicted noise $\epsilon_\phi$. By minimizing this difference, the rendered samples become more similar to the plausible samples generated by the pretrained diffusion model.

## 3.2 PSHEAD

Our goal is to generate a high-fidelity 3D head model parameterised $\theta$, that preserves the identity and appearance of the person in a frontal reference image $\mathbf{I}_{ref}$. To achieve this, we leverage prior knowledge embedded in models pretrained on both general objects and faces to optimize $\theta$ from coarse to fine. In the coarse stage, we use a mixture of SDS losses provided by personalized T2I, generic I2I, and T2V diffusion models to optimize a coarse 3D-GS. The reference view reconstruction is also involved to supervise training. In the fine stage, we utilize personalized T2I, a shape-guided face controlnet module, and a pretrained face super-resolution model to denoise and improve novel views, and subsequently apply reconstruction loss using the enhanced images to refine 3D representation. Figure 2 provides a visual diagram of PSHead, illustrating these processes.

### 3.2.1 DIFFUSION WITH MIXTURE OF SDS

To learn a 3D representation for a given face image, we split the training views into two groups and apply losses on images rendered from different viewpoints to optimize 3D-GS $\theta$. The first group consists of the reference view of the input image: $I'_{ref}$, which is supervised by a reference image $I_{ref}$, using a combination of L1 and L2 loss to measure the pixel-wise difference:

$$\mathcal{L}_{ref} = ||I'_{ref} \odot M - I_{ref}||_2^2 + ||I'_{ref} \odot M - I_{ref}||_2^1, \tag{3}$$

where $\odot M$ is a Hadamard product. We apply a foreground mask $M$ to isolate the object of interest, which helps simplify and improve the geometry reconstruction process (Yariv et al., 2020).

The second group includes novel views of the object, where we uniformly sample 25 views around the azimuth angle from $0°$ to $360°$. These novel views are optimized under the guidance of prior models to improve the overall training process and reconstruction quality.

Specifically, we investigate three types of diffusion priors: T2I (Ruiz et al., 2023), I2I Liu et al. (2023), and T2V Wang et al. (2023b). T2I focuses on how text descriptions influence individual novel view generation, I2I examines how a reference image propagates to the generation of another view, and T2V explores multi-view generation in a sequential manner. To fully harness the potential of these diffusion models, we have made several improvements and modifications, effectively conditioning them to enhance the 3D-GS process.

**Personalized Text-to-Image (T2I) SDS.** Text-to-3D avatar generation, which creates a 3D head avatar using descriptive text through SDS loss, has shown promising performance (Han et al., 2024; Chen et al., 2023; Liu et al., 2024). However, when applied directly to the task of image-to-3D generation, it often results in a mismatch between the generated 3D avatar and the identity of the reference image. This is due to the inherent ambiguity of text – a picture is worth a thousand words. To facilitate the understanding of visual characteristics of a given image, we propose combining descriptive text with a personalized T2I diffusion model.

Specifically, we utilize BLIP (Li et al., 2022) to describe a face from three key aspects: gender, clothing, and hair style. Additionally, we deploy DreamBooth (Ruiz et al., 2023), to personalize T2I model to encode reference image through few-shot tuning, which helps to reduce the excessive imagination typically seen in 2D diffusion models. To generate the necessary inputs for fintuning, we follow (Huang et al., 2024) to augment the single input image with five different backgrounds, and create a gallery of "man" and "woman" images for regularization. After optimization, the subject-specific appearance is encoded within a unique identifier token "[V]". For instance, the description for the reference face in Figure 2 is "photo of a [V] man, light blonde hair, detailed strand hair, black v-neck shirt, photorealistic, 8K, HDR." To update 3D-GS, we specify Eq 2 with T2I SDS:

$$\nabla_\theta \mathcal{L}_{SDS_{t2i}} = \mathbb{E}_{\epsilon,t}[w_t(\epsilon_\phi(I_t; y) - \epsilon)\frac{\partial(I)}{\partial(\theta)}]. \tag{4}$$

Here, $y$ represents the prompt for the reference image. However, the loss in Eq 4 optimizes each generated image separately, without explicitly enrolling the reference image, resulting in two potential issues: inconsistencies in geometry and visual appearance across different views, and generated images that may not accurately reflect the reference image. To address these problems, additional regularization is needed to ensure coherence and fidelity across all generated views.

**Image-to-Image (I2I) SDS.** We use Zero123 (Liu et al., 2023) to correlate novel views with the reference image, Zero123 is a finetuned version of image diffusion model designed for view-conditioned image generation. After being trained on synthetic 3D datasets, it has acquired rich 3D priors about the visual world. The model uses a reference image and external camera parameters as inputs, allowing it to generate novel views of the same subject while maintaining consistency with the reference image. Here, given a reference image $I_{ref}$ at $v_{ref}$ and a relative camera pose transformation $\Delta v$, we compute the SDS loss using Zero123 to update 3D-GS as follows:

$$\nabla_\theta \mathcal{L}_{SDS_{i2i}} = \mathbb{E}_{\epsilon,t}[w_t(\epsilon_\phi(I_t; I_{ref}, \Delta_v) - \epsilon)\frac{\partial(I)}{\partial(\theta)}], \tag{5}$$

where $I$ is a rendered image at $v = v_{ref} + \Delta_v$ and $I_t$ is its noised version. In our experiment, we assume the reference image corresponds to a front view.

**Text-to-Video (T2V) SDS.** To improve the consistency of multi-view images in a single batch, we employ a T2V diffusion model. Vivid123 (Kwak et al., 2024) introduced the idea of treating

novel-view synthesis as a sequential frame generation problem, innovatively combining novel-view diffusion models like Zero123 with video diffusion models. This approach effectively addresses issues such as pose inconsistencies and abrupt changes between synthesized views.

Building on this idea, we propose to leverage the temporal consistency inherent in the T2V model to ensure spatial 3D consistency across different camera viewpoints. To minimize unintended creative deviations from the text description, we employ the T2V model with a null prompt. The SDS gradient for the T2V model is expressed as follows:

$$\nabla_\theta \mathcal{L}_{SDS_{t2v}} = \mathbb{E}_{\epsilon,t}[w_t(\epsilon_\phi(\mathrm{I}_{1:T}^t; \mathrm{I}_{ref}) - \epsilon)\frac{\partial(\mathrm{I}^{1:T})}{\partial(\theta)}], \tag{6}$$

where $\mathrm{I}_{1:T}^t$ represents noised version of images rendered from a sequence of camera views. Supervision from the reference view affects all views, as they are processed as a single input to T2V. This ensures more coherent and stable 3D reconstructions across different angles.

### 3.2.2 Self-Enhancement via Image Refinement

We observe that using diffusion SDS losses for generating faces often leads to over-saturation, artifacts (see Figure 4(S3)). This occurs for two key reasons: (1) SDS loss tends to optimize for an average across different noise levels, leading to over-saturated color blocks, and (2) models like T2I, I2I, and T2V are trained on general object datasets, not face-specific ones, which limits their ability to capture facial details. Inspired by the denoising nature of diffusion models, a line of works (Zhou & Tulsiani, 2023; Tang et al., 2024; Zhu et al., 2024) have explored image-space reconstruction to address these challenges. Following this, we propose a self-enhancement module that first renders a blurry image from any given camera view I, and then reconstruct it from a clean version $\mathrm{I}^{sr}$, predicted by a denoise model. We utilize both pixel-level and perceptual losses for reconstruction:

$$\mathcal{L}_{refine} = ||\mathrm{I} - \mathrm{I}^{sr}||_2^2 + ||\mathrm{vgg}(\mathrm{I}) - \mathrm{vgg}(\mathrm{I}^{sr})||_2^2. \tag{7}$$

**Personalized T2I Refinement.** We first re-use the personalized T2I model in Sec. 3.2.1 for refinement. We add random noise on rendered image and apply a coarse multi-step denoising process $f_{t2i}$ using a personalized T2I model to obtaining a refined image:

$$\mathrm{I}_{t2i}^{sr} = f_{t2i}(\mathrm{I} + \epsilon; y). \tag{8}$$

**Landmark ControlNet Refinement.** We also incorporate a landmark-guided ControlNet (CrucibleAI, 2023) trained on a comprehensive 2D facial dataset to refine the entire image. In this case, we first utilize Mediapipe (Lugaresi et al., 2019) to detect 478 landmarks $p_{ref}$ in the reference image. We also obtain a depth map $D_{ref}$ from depth rendering of the coarse reconstruction on the reference image, and use it to obtain the depth of the landmarks and reproject them into near-frontal views from different camera angles. We then add random noise on rendered image and apply a coarse multi-step denoising process $f_{lmk}$ using landmark-guided ControlNet to obtaining a refined image:

$$p_{v_i} = K\pi_{v_i}\pi_{v_{ref}}^{-1}D_{ref}(p_{ref})K^{-1}p_{ref},$$
$$\mathrm{I}_{lmk}^{sr} = f_{lmk}(\mathrm{I} + \epsilon; p_{v_i}), \tag{9}$$

where $K$ is camera intrinsic parameters and $\pi$ refers to the extrinsic camera parameters.

This process generates shared 3D facial landmarks for each face during a training iteration, allowing a shape-guided diffusion model to produce $\mathrm{I}^{sr}$. This approach has been effective in prior works like HeadArtist (Liu et al., 2024) and HeadSculpt (Han et al., 2024) for preserving geometry consistency, and our method adds flexibility by eliminating the need for a pre-calculated head template.

**Face Super-Resolution Refinement.** We further use a face super-resolution model, CodeFormer (Zhou et al., 2022), to enhance the face facial details. By detecting the face region with RetinaFace Deng et al. (2020), we can apply CodeFormer $f_{cf}$ to predict a clean face image:

$$\mathrm{I}_{cf}^{sr} = f_{cf}(crop(\mathrm{I})). \tag{10}$$

By utilizing kornia [1], the entire align-cropping process ($crop$) becomes differentiable. Our method combines all these together and applies them at the middle point of our training. The personalized T2I is applied on the back side view to denoise hair where the face detection fails while landmark-guided ControlNet and CodeFormer are applied to other views to enhance facial regions.

---

[1]https://github.com/kornia/kornia

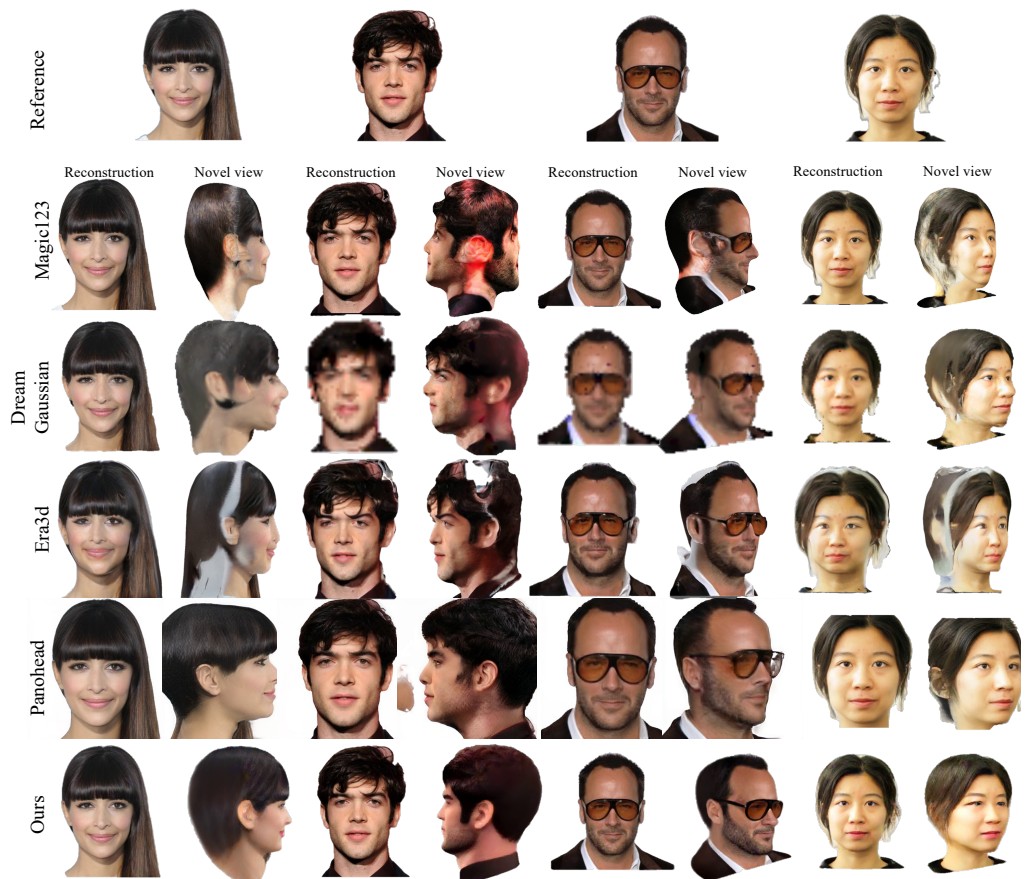

Figure 3: Qualitative evaluation of image-to-3D methods. Our approach outperforms previous methods in reconstructing the reference image and synthesizing novel views.

### 3.2.3 OPTIMIZATION

In addition, we also seek identity preserving loss between rendered images and the reference image to maintain identification. This is achieved by:

$$\mathcal{L}_{id} = 1 - cos(I, I^{ref}),$$ (11)

where $cos$ measures Arcface (Deng et al., 2019) feature cosine similarity between rendered image and reference image. We alternately optimize $\theta$ using gradients derived from different sources: image reconstruction loss (Eq 3), T2I SDS (Eq 4), I2I SDS (Eq 5) and T2V SDS (Eq 6). At the midpoint of the process, we begin refining the rendered images, using these refined images to further supervise the optimization (Eq 7 while main identity (Eq 11)improving consistency and quality.

## 4 EXPERIMENT

### 4.1 SETTING

**Dataset.** To assess our method, we establish a benchmark which includes images from PointAvatar (Zheng et al., 2023), CelebA (Liu et al., 2015) and our captured data. An effective 3D model should reconstruct reference view at reference view point while maintaining consistent semantics with the reference across different viewing angles.

**Metrics.** We evaluate these two aspects using the following metrics (Tang et al., 2023; Qian et al., 2024): PSNR and LPIPS (Zhang et al., 2018) to measure the reconstruction quality from the reference image. Contextual Distance (CD) (Mechrez et al., 2018), CLIP Similarity (CLIP) (Radford et al., 2021) and ID Similarity (ID) (Deng et al., 2019) assess the similarity between novel-view rendering images and the reference image. To consider the multi-head issues in generation, we apply a 20% penalty to the novel-view measurement when the face is visible at the backside view.

| Method | Ref view | | Novel views | | |
|---|---|---|---|---|---|
| | PSNR↑ | LPIPS↓ | CD↓ | CLIP↑ | ID↑ |
| Magic123 (Qian et al., 2024) | 27.45 | 0.028 | 2.20 | 0.57 | 0.65 |
| DreamGaussian (Tang et al., 2024) | 23.37 | 0.079 | 2.07 | 0.58 | 0.60 |
| Era3D (Li et al., 2024) | 14.43 | 0.186 | 2.08 | 0.60 | 0.58 |
| PanoHead (An et al., 2023) | 26.17 | 0.039 | 2.63 | 0.65 | 0.65 |
| Ours | **28.50** | **0.024** | **1.91** | **0.67** | **0.70** |

Table 2: Quantitative comparisons with state-of-the-art methods on single view reconstruction and novel view synthesis. The results are averaged across novel-views.

**Implementations.** During the training process, we assume the input image is captured from a frontal view, with the initial polar angle set at $90°$ and the azimuth angle at $0°$. For training new views, we uniformly sample 25 views across a full azimuth range of $360°$ while keeping the camera's polar angle fixed at $0°$. The distance from the camera to the object's center remains constant throughout the training process. Our code is implemented in PyTorch built upon threestudio [2]. Our 3D-GS are initialized with random points. We train for a total of 2000 iterations per input. The resolution is progressively increased from 128 to 256 and then to 512 at the 200-th and 300-th iterations, respectively. After 1000-th iterations, we apply image refinement. The entire optimization process takes approximately 1.5 hour on a single NVIDIA A100 (80GB) GPU. For setting hyperparameters, the guidance scales are set to 25 for T2I, 3 for I2I, and 100 for T2V. The loss function weights are set as follows: T2I $\in \{0.1, 0.5\}$, I2I $\in \{0.1, 0.5\}$, T2V $\in \{0.01, 0.1\}$, and 10 for $\mathcal{L}_{refine}$. Other configurations all follow DreamGaussian (Tang et al., 2024).

**Competitors.** We compare PSHead with four state-of-the-art methods: Magic123 (Qian et al., 2024), DreamGaussian (Tang et al., 2024), Era3D (Li et al., 2024) and Panohead (An et al., 2023). We use their official code implementations and follow their preprocessing steps.

### 4.2 RESULTS

**Qualitative Comparisons.** Figure 3 shows qualitative comparison on novel view synthesis between PSHead and its competitors. Magic123 struggles to accurately reconstruct the reference head and experiences "Janus" issues, where the avatar displays multiple inconsistent faces. Even with Zero123, it remains unclear about the correct camera view. DreamGaussian produces very blurry images and fails to generate convincing novel views with large poses, showing that I2I alone can estimate rough face geometry but lacks details. Era3D improves diffusion guidance but still generates artifacts in unseen regions and distorts the head shape, while capturing more details compared to DreamGaussian, the artifacts also become more pronounced. Panohead, trained on large-scale 2D face images, can generate novel views but encounters artifacts in the background, ears, and eyeglasses. In comparison, our method achieves remarkably faithful appearance under novel views.

**Quantitative Comparisons.** As shown in Table 2, our approach significantly outperforms the competitors in both reference-view and novel-view evaluations. Our method ranks Top-1 across all metrics when compared to state-of-the-art methods, with PSNR and LPIPS demonstrating notable improvements, underscoring superior reconstruction quality. The enhanced CLIP-Similarity indicates strong 3D coherence with the reference view. Also, our method excels in the ID-similarity, showcasing its ability to accurately capture facial features and maintain high identity consistency across different novel viewpoints. An interesting finding is that although Magic123 exhibits significant multi-head issues and distorted faces in its rendered images, as shown in Figure 3, distorting the reference appearance into other views makes it achieve high ID score.

### 4.3 ABLATION STUDIES

We conduct ablation studies to analyze the different components of proposed method PSHead with quantitative comparison in Table 3 and qualitative comparison in Figure 4.

**The effect of personalized T2I SDS.** To evaluate its impact, we run experiments without using DreamBooth to personalize the T2I model. As shown in Figure 4, this component is essential for accurately capturing facial characteristics from the reference image. Without it, the vanilla model

---

[2]https://github.com/threestudio-project/threestudio

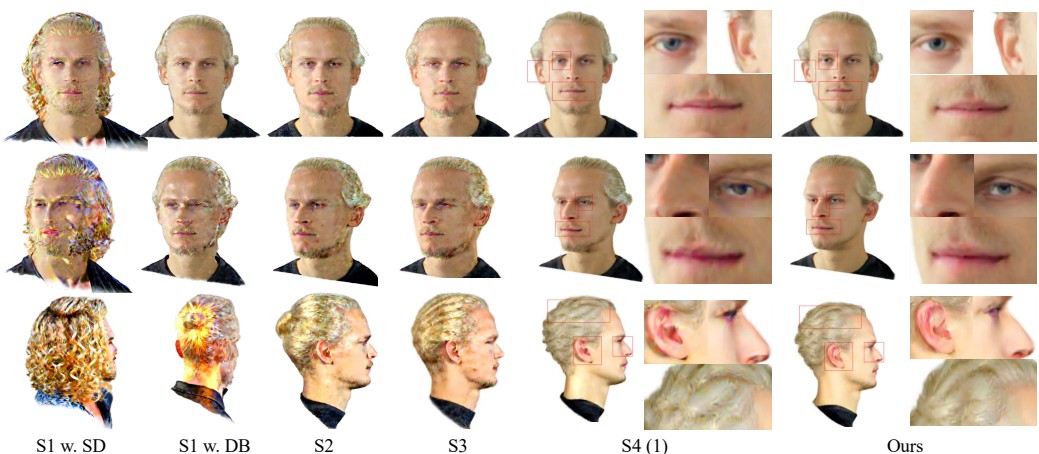

S1 w. SD          S1 w. DB          S2          S3          S4 (1)          Ours

Figure 4: Ablation study. Detailed specification are shown in Table 3. The reference image is in Figure 2. When using the personalized T2I model trained with DreamBooth, the hairstyle and identity more closely matches the reference image (S1 w.DB vs S1 w.SD). With the addition of I2I guidance (S2), the novel views align more closely with the reference image. The use of T2V reduces noise (S3), and further noise reduction is achieved through image refinement with the personalized T2I model (S4(1)). Our full method, which incorporates face-specific models for additional refinement, produces more natural and high-fidelity novel views.

| Variants | $\mathcal{L}_{SDS_{t2i}}$ SD | $\mathcal{L}_{SDS_{t2i}}$ DB | $\mathcal{L}_{SDS_{i2i}}$ | $\mathcal{L}_{SDS_{t2v}}$ | $\mathcal{L}_{refine}$ DB | $\mathcal{L}_{refine}$ CN | $\mathcal{L}_{refine}$ CF | $\mathcal{L}_{id}$ | Ref View PSNR↑ | Ref View LPIPS↓ | All Views CD↓ | All Views CLIP↑ | All Views ID↑ |
|---|---|---|---|---|---|---|---|---|---|---|---|---|---|
| S1 | ✓ | ✗ | ✗ | ✗ | ✗ | ✗ | ✗ | ✗ | 13.74 | 0.323 | 2.74 | 0.54 | 0.18 |
| S1 | ✗ | ✓ | ✗ | ✗ | ✗ | ✗ | ✗ | ✗ | 19.68 | 0.121 | 2.01 | 0.74 | 0.56 |
| S2 | ✗ | ✓ | ✓ | ✗ | ✗ | ✗ | ✗ | ✗ | 20.71 | 0.101 | 1.69 | 0.78 | 0.59 |
| S3 | ✗ | ✓ | ✓ | ✓ | ✗ | ✗ | ✗ | ✗ | 21.37 | 0.094 | 1.47 | 0.85 | 0.63 |
| S4 | ✗ | ✓ | ✓ | ✓ | ✓ | ✗ | ✗ | ✗ | 26.23 | 0.028 | 1.37 | 0.89 | 0.68 |
| S4 | ✗ | ✓ | ✓ | ✓ | ✓ | ✓ | ✗ | ✗ | 28.98 | 0.025 | 1.35 | 0.89 | 0.69 |
| S4 | ✗ | ✓ | ✓ | ✓ | ✓ | ✓ | ✓ | ✗ | 29.43 | 0.022 | 1.31 | 0.91 | 0.68 |
| **Ours** | ✗ | ✓ | ✓ | ✓ | ✓ | ✓ | ✓ | ✓ | **30.98** | **0.021** | **1.31** | **0.91** | **0.76** |

Table 3: Ablation studies on different components. The results are averaged across novel-views. DB, CN and CF denote DreamBooth, landmark-guided ControlNet and CodeFormer, respectively.

is heavily influenced by the text description, especially when reconstructing hair. For instance, the model without DreamBooth (S1 w. SD) tends to generate long blonde hair, whereas the version with DreamBooth (S1 w. DB) is able to replicate the hairstyle from the reference image.

**The effect of I2I SDS.** By adding the I2I SDS loss (S2), we observe a substantial improvement over S1 w. DB at large view points. S1 w. DB generates distorted faces at large angles because it relies on rough, sparse view descriptions like front, back, and side to represent view angles. In comparison, I2I SDS loss built upon Zero123 more effectively propagates the reference view to novel angles through conditioning on a more precise camera pose transformation.

**The effect of T2V SDS.** Our findings slightly differ from Vivid123 (Kwak et al., 2024). While Vivid123 reports that T2V reduces abrupt view changes in novel view synthesis, we did not observe such changes without it when optimizing 3D-GS for face using SDS loss. However, we found that in some cases, T2I and I2I face multi-head issues, while T2V successfully rotates the object, producing better results (See Figure 9 in Sec. 7). Besides, incorporating T2V led to smoother images in quantitative results, so we add it to increase model's generality when handling diverse inputs.

**The effect of Self-Enhancement.** S4(1) integrates image enhancement through personalized T2I, leading to a noticeable reduction in artifacts compared to models without this enhancement. Notably, these results already outperform image-to-3D diffusion-based competitors discussed in Sec. 4.2, underscoring the effectiveness of our modifications. This highlights the success of our approach in refining and optimizing existing techniques, further advancing the state of the art in this domain. However, artifacts in the face regions remain. To address this, we retain the hair region for refinement with personalized T2I, while progressively adding further refinements using landmark-guided ControlNet and CodeFormer. Our results show that each component independently contributes to improving the generation quality, with the combined use of all elements producing the most effec-

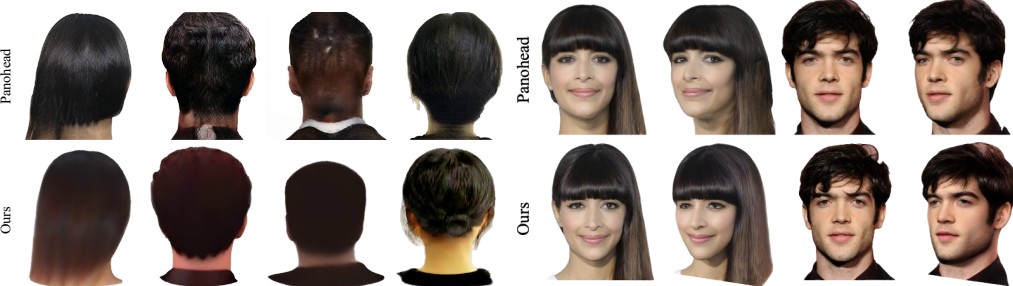

Figure 5: Comparison between without and with landmark-guided ControlNet.

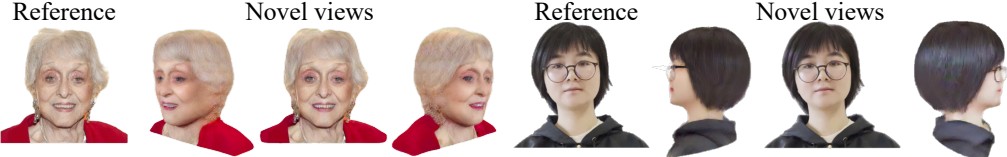

(a) Backside view of different hair styles.    (b) Gaze directions at front view and side view.

Figure 6: Comparison between Panohead and ours on different hair styles at back side view, and gaze direction in synthesized novel views. Reference images are in Figure 3.

tive outcomes. We compare with and without landmark-guided ControlNet in Figure 5, showing smoother skin and reduced noise around the eye region after introducing landmark-guided Control-Net.

Figure 7: Failure cases on reconstructing accessories like earrings and transparent eyeglasses.

## 5 MORE ANALYSIS

We conduct more comparison with Panohead, a strong competitor in Sec. 4.2. **(1) Face size.** Panohead being trained on aligned and cropped faces, struggles to handle shoulders and requires identical preprocessing during testing. In contrast, our method, built on diffusion models pretrained on a large-scale object dataset, effectively handles variations in the upper body (see Figure 1). A detailed comparison can be found in Figure 10 in Sec.7. **(2) Back side view.** Benefiting from training a discriminator on large-scale 2D hair images, Panohead generates higher-quality hair compared to our method in terms of individual hair strands. However, our method adapts better to a variety of hairstyles, whereas Panohead struggles with unseen hairstyles and tends to produce artifacts, especially on the backside (see Figure 6[a]). **(3) Gaze direction.** Since Panohead is trained on 2D face images primarily captured in controlled settings with the subject facing the camera, it often generates images where the gaze is directed straight ahead. In contrast, our method generates more natural gaze variations that adjust to different viewing angles during rendering (see Figure 6[b]).

## 6 CONCLUSION AND DISCUSSION

We propose PSHead that utilizes diffusion priors via SDS to generate coarse representation for a single reference image, which is then refined using facial priors to enhance the rendered images. Benefits from general diffusion, PSHead is robust across varying face sizes. As the first effort to integrate T2I, I2I, and T2V diffusion models into a single framework, we analyze the function of each model, hoping to inspire future work to adopt similar designs. While PSHead improves performance and expands the scope of 3D avatar generation, it has certain limitations. Due to the lack of a hair super resolution module, usage of T2I model results in the hair details appearing less defined and lacking individual strands. Additionally, accessories like earrings and transparent eyeglasses are difficult to synthesize (see Figure 7).

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

# 7 APPENDIX

## A MORE EXPERIMENT DETAILS

**Hyperparameters.** Due to the diversity of input images, finding consistent hyperparameters is challenging. Some empirical tips are: use T2I 0.5, I2I 0.1, and T2V 0.01 for default. If the generated views show over-saturation in hair, reduce T2I from 0.5 to 0.1. If the multi-head issue arises, increase I2I to 0.5, and if it persists, raise T2I to 0.1.

**Metrics.** 1) PSNR measures the reconstruction quality from the reference image at the pixel level; 2) LPIPS assesses reconstruction quality from the perceptual generation quality at the reference image. 3) Contextual Distance (CD) assess the similarity of textures between novel-view rendering images and the reference image; 4) CLIP Similarity (CLIP) assess the similarity of semantics between the novel-view rendering images and and the reference image; 5) ID Similarity (ID) computes the average cosine similarity score using ArcFace across viewpoints ($-45°$ to $45°$) relative to the reference image. If facial landmarks are undetectable in an image from a particular viewpoint, its score is set to 0. The ArcFace model used here differs from the one in Eq.11, as we use ResNet50 trained on WebFace600K, whereas the model in Eq.11 is ResNet100 trained on MS1MV2. ArcFace model used here is different from Eq. 11 (ResNet50@WebFace600K vs ResNet100@MS1MV2).

**More ablation studies.** Figure 8 shows the comparison between results without and with personalized T2I. As discussed in Sec. 3, T2I introduces creativity in generating novel views. This figure serves as visual evidence of that. Without personalized T2I, the backside view of the generated 3D representation is not necessarily incorrect, but it lacks a specific hairstyle. In contrast, personalized T2I uses its imagination to add style while staying faithful to the reference image. For instance, when the reference image features an updo hairstyle, the model generates a rounded bun at the back in the 3D view.

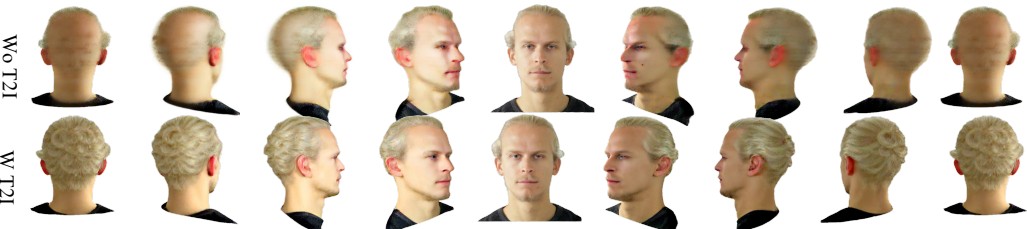

Figure 8: Comparison between without and with Personalization T2I.

Figure 9 shows the comparison between results without and with I2V. In some cases, I2I alone fails to predict the backside view of an input image, leaving a hole in the 3D-GS. In such cases, the backside is rendered using Gaussians from the front, leading to multi-head issues. Adding I2V alleviates the challenge of predicting the backside view by estimating a temporal transformation of the input image from front to back, improving the overall consistency and reducing errors in the backside generation.

Figure 10 shows Panohead results when the shoulder is included in the input. While the align-cropping process effectively focuses on the head region, it inevitably includes parts of the shoulder and clothes. This leads to Panohead generating distorted backside views around the neck and shoulder, as it is not well-suited for handling these additional regions.

## B ETHICS STATEMENT

Our proposed method, PSHead, for generating 3D avatars from a single image holds great potential to drive metaverse development forward, but it also raises concerns about possible misuse. The relative ease of obtaining personal and detailed single images, compared to multi-view images, increases the risk of malicious applications.

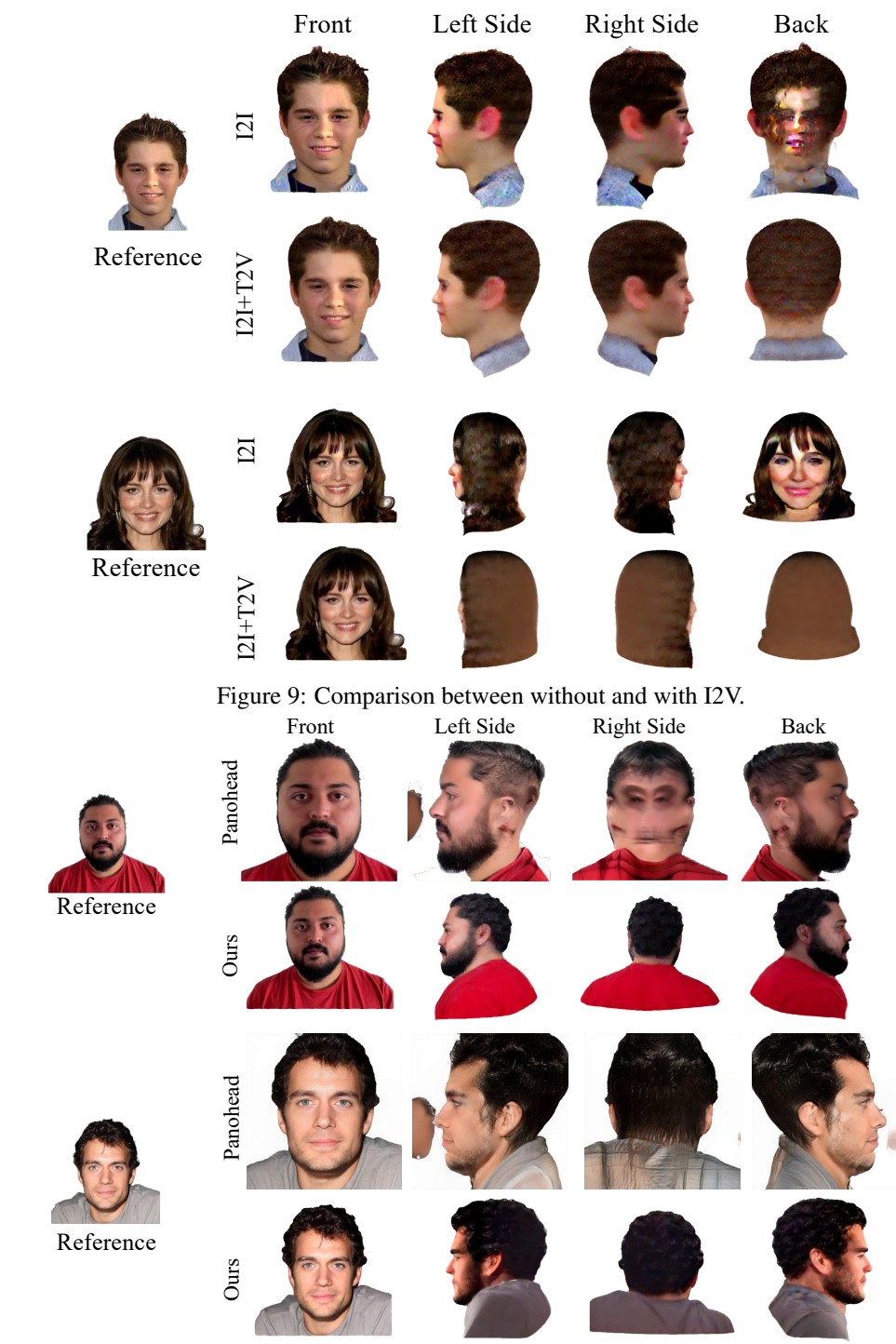

Figure 9: Comparison between without and with I2V.

Figure 10: Panohead struggles when shoulder is visible.

## C  CODE AND VISUAL RESULTS.

**Code** is in the code folder. **Video results** is in the result folder. More comparison in `https://drive.google.com/drive/folders/1-nCbi1NJoSCv13V7hhKxmacnCajkap54?usp=sharing`

