# OpenReview forum: "PSHead: 3D Head Reconstruction from a Single Image with Diffusion Prior and Self-Enhancement"
_ICLR.cc/2025/Conference — ICLR 2025 Conference Withdrawn Submission_

### Official Review · Reviewer_8AKB · 2024-10-24

**Soundness:** 3
**Presentation:** 3
**Contribution:** 2
**Rating:** 5
**Confidence:** 4

**Summary:**

The paper introduces an approach to head generation from a single image. The generation process consists of two stages: multiple pretrained models mixed SDS initialization and head-specific refinement. The framework results in realistic
$360^{\circ}$ head rendering.

**Strengths:**

1. The proposed PSHead follows the pipeline of DreamGaussian, which also includes SDS-based initialization and image-based refinement. The author(s) add several well-designed components, such as DreamBooth, T2V-SDS, and Landmark ControlNet, improving the head generation quality compared to the baseline.
2. The paper includes comprehensive experiments to evaluate the effectiveness of each design.

**Weaknesses:**

- In the introduction, the author emphasizes that "the normalization preprocessing steps of existing methods struggle in handling cases with varying scales." However, the results of PanoHead in Fig. 3 (also with shoulders) do not seem that bad. If this is a main motivation, I suggest conducting more comparisons to support it in the main paper rather than only showing a few cases in the appendix.
- The results in Fig.2 are not satisfactory, with apparent appearance and shape consistency. The $360^\circ$ videos in supplementary also show severe blur in novel views, especially in back views. In comparison, the results of Panohead are more realistic. It will be better to provide a more detailed analysis of these issues, including potential causes and ideas for improvement.
- The added SDS strategy and refinement components lead to severe efficiency degradation, nearly 1.5 hours as reported in implementations. I think the authors should conduct a comparison of runtime vs. quality metrics analysis of the trade-offs between quality improvements and computational cost.
- In ablations (section 5), the observation of the gaze direction is intuitive. However, it seems that there are no similar issues in Fig.3.

**Questions:**

- Although the multi-modality (text, image, and video) SDS work, I am not that confident of the motivation. Is it not enough with a single image-to-video model? An analysis or an ablation is necessary.
- The GS representation usually results in the degradation of geometry. I hope for more geometry comparisons with Ponahead.

**Details Of Ethics Concerns:**

No need ethic review.

---

### Official Review · Reviewer_P7aq · 2024-10-25

**Soundness:** 3
**Presentation:** 3
**Contribution:** 2
**Rating:** 3
**Confidence:** 5

**Summary:**

This paper investigates the problem of creating high-fidelity photorealistic 3D avatars from only a single face image. They propose a method that learns a 360◦ 3D-GS representation for a reference image with varying face sizes, leveraging a mixture of diffusion priors to generate a coarse representation and refine the coarse representation in an innovative way by introducing 2D face priors.

**Strengths:**

The proposed method demonstrates impressive results in generating high-fidelity photorealistic 3D avatars from a single-face image. The use of a 360◦ 3D-GS representation allows for capturing detailed facial features.

**Weaknesses:**

1. The paper claims to have achieved great 360 free view rendering. However, upon examining the visual results in the paper, it can be observed that the side view and back view contain excessive noise and are significantly blurrier than the front view. In comparison, it does not appear to be better than PanoHead.

2. Many techniques employed in this paper have been used in other papers with similar goals, but they don't address the limitations of these techniques. For example, in the refinement stage, it is unclear how the multi-view inconsistency of refined novel views is handled.

3. Mixed SDS. This paper utilizes three types of SDS loss. However, in Figure 4, it seems that T2V SDS only provides marginal enhancements compared to I2I SDS. Although improvements are shown in Table 3, it is not demonstrated whether T2V still performs well when the refinement stage is followed by only T2I + I2I.

4. The method section indicates that the geometry is primarily based on the SDS loss. While personalized diffusion models are mentioned, it remains unclear whether the geometry captures intrinsic details and performs better than generic SDS methods.

5. The paper reports better numerical results for novel views compared to the comparison methods. However, it is worth noting that most metrics for evaluating novel views are done in feature space rather than pixel space(such as psnr). This could explain why the novel views generated by this method appear blurry, but still achieve higher scores than the baselines.

6. The preservation of identity in the rendered avatars from novel views appears to be weak, as observed in Figure 3. In column 4, there is a noticeable change in identity.

**Questions:**

1. Questions: I don't have too much question for this paper.
2. Suggestions: It is recommended that the authors focus on improving the quality of the side view and back view in order to achieve better results. Additionally, they should validate the effectiveness of using mixed SDS loss by comparing it with one or two SDS loss that can potentially achieve similar performance when combined with the refined stage. Furthermore, conducting evaluation in pixel space for novel views would provide more comprehensive results.

---

### Official Review · Reviewer_i9LZ · 2024-10-25

**Soundness:** 2
**Presentation:** 3
**Contribution:** 1
**Rating:** 3
**Confidence:** 5

**Summary:**

- This paper leverages the human face priors (e.g.,   Face landmarks and Face ID) and numerous 2D diffusion models via SDS to establish a coarse-to-fine pipeline for generating 3D avatars from a single image.
- The proposed method consistently surpasses existing techniques (Magic123, DreamGaussian, Era3D, and PanoHead) on PointAvatar, CelebA, and a private dataset, achieving superior quantitative and qualitative results.
- Detailed results and corresponding code are included in the supplements.

However, the technical novelty is limited, as it primarily uses existing modules, and the empirical approaches for generating 3D Head Avatars from single images are typical.

**Strengths:**

**1.** It includes a comprehensive review of related works.

**2.** The work effectively integrates existing modules and validates the efficacy of critical design components. Furthermore, it addresses a significant problem in the field of 3D Head Reconstruction.

**Weaknesses:**

**1.** The work presents incremental methods, mainly refining Head Reconstruction with predictable improvements and relying extensively on off-the-shelf modules such as 2D pre-trained diffusion models, face landmark detection, and ID recognition model (Arcface) for loss function. Specifically：
  - Coarse stage: Employs DreamBooth for personalized T2I diffusion to produce a preliminary 3D-GS.
  - Fine stage: Utilizes personalized T2I diffusion, landmark-guided ControlNet, and a pre-trained face refinement model (CodeFormer) .

The authors should discuss the design intuition rather than empirically constructing an engineering pipeline.

**2.** The complexity of the engineering pipeline, detailed in **Figure 2** and **Section 3**, makes the work hard to follow and may hinder further exploration and industrial applications.

The authors should reduce the number of modules, focusing on core modules as the main claim.

**3.** PSHead lacks the capability to drive expressions.

Unlike previous works such as HeadGAP and Morphable Diffusion, PSHead does not support expression-driven animation, limiting its applicability to various downstream applications.

**4.** The paper omits crucial information about model parameters and reconstruction times compared to cutting-edge 3D generation works (e.g., in **Tables 2** and **Table 3**).

**5.**  The per-instance optimization process takes approximately 1.5 hours (refer to Implementations), indicating high computational demands.


I would appreciate it if the authors could address my concerns by providing corresponding quantitative or qualitative results based on the **weaknesses** and **review feedback**.

**Questions:**

- As depicted in **Figure 10**, is PSHead capable of effectively managing tasks involving the reconstruction of the head, upper body, and full body?
- Does PSHead exhibit any racial inductive biases?

---

### Official Review · Reviewer_9agS · 2024-11-03

**Soundness:** 2
**Presentation:** 2
**Contribution:** 2
**Rating:** 5
**Confidence:** 4

**Summary:**

This paper introduces a new approach called PSHEAD for generating high-quality 3D avatars from a single image. The key contribution of this research is the utilization of a mixture of diffusion priors to create a coarse representation of the input face, which is then refined through the integration of 2D face priors. Experiments demonstrate promising results, outperforming several baselines.

**Strengths:**

1. The paper successfully demonstrates the effectiveness of integrating T2I, I2I, and T2V diffusion models into a single framework for generating 3D avatars, showing good performance.
2. The paper is  well-written and easy to follow.
3. The experimental results demonstrate better performance than the baselines in single-view reconstruction.

**Weaknesses:**

1. My main concern lies in the technical contributions of this paper. The authors combine multiple models, such as T2I, I2I, and T2V, to achieve state-of-the-art results. They should provide more insights regarding the use of these models in the paper.
2. The author should explain why the I2V model was not used and include an ablation study for the I2V model.
3. The optimization-based method takes a long time to create a human head Gaussian model, requiring approximately 1.5 hours on a single NVIDIA A100 (80GB) GPU, which makes it difficult to use in practical applications.

**Questions:**

1. Can the generated head Gaussian model be driven? If so, please illustrate some novel pose synthesis results.
2. Missing some references:
    [1] AvatarCLIP: Zero-Shot Text-Driven Generation and Animation of 3D Avatars;
    [2] DreamHuman: Animatable 3D Avatars from Text;
    [3] TADA! Text to Animatable Digital Avatars;
    [4] ZHOU Z., MA F., FAN H., YANG Y. Headstudio: Text to animatable head avatars with 3d gaussian splatting.
3. In the ablation study, as Fig. 4 and Tab. 3 show, self-enhancement plays an essential role in generating quality outputs. Does this mean that you do not require the all of diffusion model priors, but that relying on a single diffusion prior, such as T2I combined with self-enhancement, is sufficient? Please provide additional ablation studies, such as T2I + self-enhancement, I2I + self-enhancement, and T2V + self-enhancement. I want to be certain that it is necessary to employ the all of diffusion model prior to distilling the initial head Gaussian model.

---

### Note · Authors · 2024-11-12

I have read and agree with the venue's withdrawal policy on behalf of myself and my co-authors.